# High Prevalence of Glaucoma among Patients in an Onchocerciasis Endemic Area (Mahenge, Tanzania)

**DOI:** 10.3390/pathogens11091046

**Published:** 2022-09-14

**Authors:** Juliet Otiti-Sengeri, Blair Andrew Omaido, Dan Bhwana, Damalie Nakanjako, Malik Missiru, Musa Muwonge, Luis-Jorge Amaral, Bruno P. Mmbando, Robert Colebunders

**Affiliations:** 1Department of Ophthalmology, College of Health Sciences, Makerere University, Kampala P.O. Box 7072, Uganda; 2National Institute for Medical Research, Tanga Centre, Tanga P.O. Box 5004, Tanzania; 3Department of Internal Medicine, College of Health Sciences, Makerere University, Kampala P.O. Box 7072, Uganda; 4Department of Ophthalmology, Mahenge District Hospital, Mahenge P.O. Box 4, Tanzania; 5School of Medicine, Soroti University, Soroti P.O. Box 690, Uganda; 6Global Health Institute, University of Antwerp, Kinsbergen Centrum, Doornstraat 331, 2610 Antwerp, Belgium

**Keywords:** onchocerciasis, epilepsy, nodding syndrome, visual acuity, glaucoma, prevalence, Tanzania

## Abstract

Onchocerciasis is known to cause skin lesions and blindness, but there is also epidemiological evidence that onchocerciasis is associated with epilepsy, including nodding syndrome. We carried out ocular exams in persons with epilepsy in Mahenge, an onchocerciasis endemic area with a high prevalence of epilepsy in Tanzania. We recruited 278 consecutive persons with epilepsy attending the epilepsy clinic at Mahenge hospital and satellite clinics in rural villages. They underwent a general physical and a detailed ocular examination and were tested for onchocerciasis Ov16 IgG4 antibodies. Glaucoma was defined by a raised intraocular pressure above 21 mmHg with evidence of typical glaucomatous disc changes in one or both eyes. Among the 278 participants, median age 27 (IQR 21–38) years, 55.4% were female; 151/210 (71.9%) (95% CI: 65.3–77.9) were Ov16 positive. The most frequent ophthalmic lesions were glaucoma (33.1%), vitreous opacities (6.5%) and cataracts (2.9%). In multivariate analysis, glaucoma (adjusted IRR = 1.46; 95% CI: 1.24–1.70) and age (adjusted IRR = 1.01; 95% CI: 1.01–1.02) were significantly associated with onchocerciasis. In conclusion, a high prevalence of glaucoma was observed among Ov16 positive persons with epilepsy. Persons with epilepsy with *O. volvulus* infection should undergo screening for glaucoma to prevent one of the causes of preventable blindness.

## 1. Introduction

Onchocerciasis (River blindness) is a neglected tropical disease caused by *Onchocerca volvulus*; a filarial nematode transmitted through bites of infected *Simulium* blackflies. The disease is the second leading cause of preventable blindness after trachoma and is endemic in populations near fast-flowing rivers where the blackflies breed [1].

A parasitised female blackfly introduces filarial larvae in their third stage onto the host’s skin during a blood meal. The larvae penetrate into the bite wound where they develop, over 6–12 months, into adult filariae, forming nodules in subcutaneous connective tissues, and can live there for up to 15 years. Nodules can contain male and female worms [2]. Male worms measure 1.9–4.2 cm in length and 130–210 micrometres in diameter, while female worms measure 33–50 cm by 270–400 micrometres. Fertilised adult female worms can produce up to 1000 microfilariae per day for about nine years, resulting in millions of microfilariae residing in the subcutaneous tissues [3,4]. The microfilariae live for 1–2 years. Consequently, when a female blackfly bites a human host, the microfilariae are transferred from the host to the blackfly, where they develop into infective larvae over 6–12 days [2].

The microfilariae migrate throughout the body, especially to the skin and eyes [5]. Most onchocerciasis clinical manifestations occur due to the direct effects of microfilariae and the host’s immune response to the microfilarial antigens [4]. Skin changes include papular onchodermatitis, hyperpigmentation, lichenified onchodermatitis, pruritus and common secondary infections. Ocular manifestations occur when microfilaria move from the conjunctiva through the cornea into the anterior and posterior chambers of the eye, leading to conjunctivitis, sclerosing keratitis, uveitis, chorioretinal lesions, optic atrophy, and glaucoma. In the most severe cases, blindness can result. Dead microfilariae in the cornea induce a tissue reaction to produce the characteristic “snowflake” opacities [6].

Approximately 218 million people worldwide are at risk of onchocerciasis. Of those, 99% live in 31 endemic countries in Sub-Saharan Africa. In Tanzania, approximately 6.5 million people are at risk of onchocerciasis [2]. Mahenge is the most onchocerciasis endemic area in Tanzania [7]. Community-directed treatment with ivermectin (CDTI) has been implemented to control disease transmission [8]. However, vector control strategies to eliminate the *Simulium* flies have not been implemented, potentially because of the high cost of vector larviciding and the numerous hard-to-access fast-flowing rivers in the area that preclude methods like “Slash and Clear” [9]. The abundance of flies predisposes the local communities to reinfection [9]. Recent serological and entomological studies suggest high ongoing onchocerciasis transmission in the rural villages of the Mahenge [10]. Consequently, the national neglected tropical diseases programme increased the frequency of CDTI in the area from annual to biannual starting in 2019 [11].

In a study in 2018, a high prevalence of epilepsy (2.7%) was observed in rural villages in the Mahenge area [8]. Nodding syndrome was observed in 12.7% of the persons with epilepsy [8]. Several studies have demonstrated an association between onchocerciasis and epilepsy including nodding syndrome [12]. However, the pathophysiology of developing epilepsy and nodding syndrome is not yet completely understood [3,13,14].

Ocular complications of onchocerciasis have been attributed to the direct infection with microfilariae, the toxic immune reactions associated with their death and the probable autoimmune reactions involving cross-reacting antigens and the interphotoreceptor retinoid-binding proteins [15,16]. There are limited studies documenting the effect of ocular onchocerciasis on intraocular pressure and the development of glaucoma. Pathology in infected eyes shows that the post-trabecular outflow system is affected in subjects with both glaucoma and onchocerciasis infection. Microfilariae infiltrate the cells around the Schlemm’s canal, the efferent veins, the episclera, and the vessels in Tenon’s capsule. This inflammation raises resistance to outflow of aqueous humour, intraocular pressure, and development of glaucoma. Long-term inflammatory responses have been documented to cause ocular hypotony [17].

Ultimately, the microfilarial-induced damage to the ocular tissues results in visual impairment and blindness. Blindness has been shown to lead to premature death due to the negative social and economic consequences [5]. Routine ocular exams should be performed in onchocerciasis endemic regions in order to identify and treat ocular diseases early and prevent irreversible blindness.

In this study, we measured intraocular pressures and determined the prevalence of glaucoma among persons with epilepsy attending a neurology clinic in Mahenge hospital and satellite treatment sites in rural villages in the area. Other ocular abnormalities were also documented. We hypothesised that having onchocerciasis may be a risk factor for developing raised intraocular pressure and glaucoma in this community.

## 2. Materials and Methods

This study implemented a cross-sectional design in Mahenge, a rural area in Tanzania.

### 2.1. Study Site and Population

The study was conducted at the epilepsy clinic at Mahenge hospital and satellite treatment sites in four villages (Mdindo, Msogezi, Mzelezi and Sali) in the Mahenge area of the Ulanga district, Morogoro region, in south-eastern Tanzania (Figure 1). Mahenge extends into a mountainous area above the Kilombero river valley, where most of the northern Ulanga district villages lie. The Ulanga district has 59 registered villages that are endemic for onchocerciasis [11].

Mahenge has a population of approximately 151,001 and a growth rate of 4.9% [18]. The area topography comprises mountainous and fast-flowing rivers and streams that provide suitable breeding habitats for the blackfly vectors of onchocerciasis [18]. CDTI has been implemented for more than 20 years. However, due to the vector abundance, reinfection is common. In 2017, the prevalence of *O. volvulus* antibodies among children 6–10 years old in rural villages in the Mahenge area was 42.6%, suggesting high ongoing onchocerciasis transmission. Based on this finding, the onchocerciasis elimination programme in Mahenge was switched from annual to bi-annual CDTI.

### 2.2. Study Subjects

A total of 278 persons with epilepsy, including nodding syndrome, were consecutively recruited as they attended their routine care at the neurology clinic at Mahenge hospital and satellite treatment sites within the region. Some patients were treated with anti-seizure medication, while others were attending the clinic for the first time.

Preverbal children and patients with extreme forms of altered mental states were excluded because they could not adequately participate in vision assessment and ocular examination.

### 2.3. Study Procedures

#### 2.3.1. Interview and Physical Examination

All persons with epilepsy were interviewed and examined by the resident physician (DB) and a team of trained nurses and clinical officers. This procedure with patients included taking their relevant medical and socio-demographic history, and documenting their weight, height, and general mental status. Inspection and identification of skin lesions, palpable lymph nodes, and stage of development for adolescents were then performed. The type of epilepsy, nodding syndrome or another form of epilepsy, was determined, and medications used were recorded. Nodding syndrome was defined according to the international consensus definition.

#### 2.3.2. Onchocerciasis Diagnosis

An Ov16 Elisa test to detect antibodies against the Ov16 antigen was used to diagnose onchocerciasis exposure. The Ov16 antigen is present in all stages of the lifecycle of *O. volvulus* and has been recommended by the WHO to detect exposure to the filarial parasite [1,19,20].

#### 2.3.3. Ophthalmological Examination

All participants underwent a full ophthalmology assessment performed by the principal investigator (JO, ophthalmologist), the resident ophthalmologist (MM) and a trained ophthalmic clinical officer (MM). This process included the following:1.Visual acuity assessment

Visual acuity was assessed using a Snellen chart placed at 6 m in a well-lit-up area. Each eye was examined separately, and an illiterate E chart was used for young children who could not read. Confrontation visual field testing was done and documented. A pinhole was used when vision was less than 6/9 to rule out a refractive error. Vision loss was defined according to the standard classification [21].


2.Intraocular pressure measurement


All patients had their intraocular pressure measured using an I Care handheld tonometer. Corneal pachymetry was performed using a handheld pachymeter, and the adjusted intraocular pressures were recorded.


3.Slit-lamp examination


The anterior chamber angle was assessed using the Van Herrick method using a Haag Streit 900 slit-lamp at the main hospital and a portable slit-lamp at the rural units. The cornea, lens and pupil functions were examined, and any abnormalities were recorded.


4.Fundus examination


The fundus was examined using an indirect ophthalmoscope after pupil dilatation with a mixture of 0.4% tropicamide and 10% phenylephrine. Disc photographs for detailed disc assessment for glaucoma were taken where indicated using a 20 D lens, a peek retinal device and a mobile phone camera.


5.Other investigations


An ultrasound B’ scan was done to rule out vitreous opacities and other posterior segment abnormalities in cases where hazy media obscured direct visualisation.


6.Diagnosis/Diagnostic criteria


Our study was placed in resource-limited settings where access to gold standard test tools was not possible. Tests such as gonioscopy, visual fields, and optical coherence tomography (OCT) could not be performed to confirm some findings. The gold standard for diagnosis of open-angle glaucoma includes an open angle viewed at gonioscopy and typical structural changes of the optic disc on photography and optical coherence tomography and evidence of progressive visual field changes using standard white-on-white perimetry. This study defined open-angle glaucoma as intraocular pressure above 21mmHg in the presence of optic disc cupping.


7.Treatment


Appropriate treatment was given to all patients when indicated. This included anti-glaucoma drops for those with raised intraocular pressure, anti-inflammatory eye drops for those with allergies and uveitis and topical/oral antibiotics for those with infections. Seizures were managed with anticonvulsant medication. Those requiring surgical intervention, such as cataract surgery, were referred and managed by the local ophthalmologist.

### 2.4. Data Management and Analysis

Data were recorded on printed forms, and all data were double entered in Excel and exported to STATA 15 for analysis. Continuous variables were summarised as mean/median. Categorical variables were summarised as percentages. The two-sample Wilcoxon test was applied for explorative data analyses to evaluate differences between groups and chi-square for association with categorical variables. A modified Poisson model with robust standard errors was used for the multivariate analysis.

## 3. Results

### 3.1. Study Population Characteristics

A total of 278 persons with epilepsy ranging in age from 4 to 81 years (median age 27 years; 55.4% females) were enrolled in the study (Table 1). The majority were peasants (e.g., farmers and/or livestock holders) residing close to a fast-flowing river. Nodding syndrome was present in 18% of the study population, and 151 (71.9%) of the 210 persons with epilepsy tested were Ov16 positive (Table 1).

### 3.2. Ophthalmological Findings

One hundred forty-five (52.2%) had normal eyes (Table 2). The most frequent ophthalmic lesions were glaucoma (33.1%), followed by vitreous opacities (6.5%) and cataracts (2.9%). Seven patients had typical onchocercal corneal opacities; five were also diagnosed with glaucoma, and two had corneal lesions alone. Other lesions suggestive of previous O. volvulus infection included 18 patients with vitreous opacities and two with optic atrophy and chorioretinal atrophy.

Of the 151 Ov16 positive subjects, 43% (95% CI: 35.1–51.0) had a raised intraocular pressure (Table 3). On average, Ov16 positive participants with glaucoma were older than Ov16 negative participants (Figure 2).

### 3.3. Factors Associated with Onchocerciasis

The Ov16 positive persons with epilepsy were older (mean age 33.9 years, SD 14.1) than those who tested negative (mean age 24.6 years, SD 11.7) (*p*-value < 0.001) (Table 3). In bivariate analysis, no association was found between body mass index, sex, nodding disease syndrome and Ov16 test result.

In multivariate analysis, glaucoma and age were significantly associated with onchocerciasis. Those with glaucoma were 46% more likely to be Ov16 positive (adjusted IRR = 1.46; 95% CI: 1.244–1.70) (Table 4).

## 4. Discussion

Raised intraocular pressure was the most often described onchocerciasis-related ocular clinical manifestation. Findings are similar to the results from previous studies in onchocerciasis endemic areas in Ghana, where individuals with glaucoma were three times more likely to test positive for onchocerciasis [22,23]. In the Nord Kivu province, an onchocerciasis hyperendemic tropical rain forest area in the Democratic Republic of Congo (DRC), 39 (1.81%) out of 2150 subjects had onchocerciasis-related eye lesions and 4 (0.19%) were blind. Chorioretinitis (0.88%) was the most frequent onchocerciasis lesion followed by keratitis (0.46%), microfilaria in the anterior chamber (0.28%), iridocyclitis (0.28%), secondary glaucoma (0.19%), complicated cataract (0.19%) and optic atrophy (0.19%) [24]. Moreover, during a screening of 2234 Sierra Leonean primary and secondary school students in an onchocerciasis endemic area before CDTI was introduced (in 1983), students with an intraocular pressure higher than 21 mm showed a higher percentage of positive skin snips than those with an intraocular pressure of 21 mm or less [25]. In contrast, there is an ophthalmological study among residents of the Yanomami Tribe in the northern Amazon, Brazil, an onchocerciasis endemic area. Of the 83 natives examined, a high prevalence of probable onchocerciasis-related eye lesions was detected, including punctate keratitis (41%), microfilariae in the anterior chamber (39%), chorioretinitis (7.2%) and anterior uveitis (6.0%), but no glaucoma [26]. However, two individuals with microfilariae in the eye had intra-ocular pressure readings significantly above average. The apparent absence of glaucoma may be related to the small sample size, but also to the lack of anterior eye inflammation because of limited ivermectin exposure. The difference between this study and the African studies may also be related to strain differences between American and African *O. volvulus* strains [27,28].

It is still unclear whether certain *O. volvulus* strains may induce different ocular pathology. In a study in three ecological zones in Nigeria between 1998 and 1999, anterior segment onchocercal lesions, punctate and sclerosing keratitis were the predominant features of the infection in the savanna zone (14.1% and 6.3%, respectively), while posterior segment lesions were much more common in the forest zone [29]. It has been suggested that blindness is more prevalent in savannah than in forest onchocerciasis areas. However, a recent meta-analysis did not support the existence of a savannah-blinding strain of onchocerciasis in West Africa [30].

Patients with onchocerciasis have more peripheral anterior synechiae due to the inflammatory reactions to microfilariae, and this could cause increased resistance to aqueous humour outflow [23]. The high prevalence of glaucoma reported in our study among individuals with epilepsy, most of them with onchocerciasis-associated epilepsy, indicates that glaucoma may be a significant contributor to blindness among people who have been *O. volvulus* infected.

One of the limitations of our study is that we did not include persons without epilepsy. Therefore, our results are not generalisable for all persons with an *O. volvulus* infection. Moreover, we did not perform skin snip testing to determine whether the participants still presented an active onchocerciasis infection. Skin snip testing was not done because bi-annual CDTI had been implemented in the Mahenge area since 2019 and most persons with epilepsy had received ivermectin in the months prior to the study. The young age of our study population and the past ivermectin exposure explains the fact that in none of them were microfilariae found in the anterior chamber and very few persons presented characteristic onchocerciasis corneal (punctuate keratitis), nervus opticus and retinal lesions [31]. Indeed, such lesions are mainly observed in older persons after prolonged *O. volvulus* exposure without ivermectin treatment [32]. CDTI has drastically reduced the threat of blindness in onchocerciasis endemic foci. This was shown in a study in Kaduna state, Nigeria, an onchocerciasis endemic area where after 27 years of CDTI, the prevalence of blindness dropped from 4.9 to 0.96%. In 2016, the most common causes of blindness were cataracts (55.2%) and optic atrophy (27.6%), whereas the most common causes in 1989 were onchocerciasis (28.3%), glaucoma (17.4%) and cataract (10.9%). People with optic atrophy were more likely to have taken fewer doses of ivermectin over the years. In 2016, no case of anterior onchocerciasis was seen among the visually handicapped, and only one individual (two eyes) with posterior onchocercal chorioretinitis. This contrasts with the finding in 1989 in which onchocerciasis, especially the posterior variety, was responsible for about one-third of the cases of blindness [33].

A limitation of our study is also that it was a cross-sectional study. Therefore, we cannot confirm the causal relationship between onchocerciasis and glaucoma nor evaluate the effect of treatment of this onchocerciasis-associated glaucoma.

## 5. Conclusions

A high prevalence of glaucoma was observed among persons with epilepsy with onchocerciasis in the Mahenge area. We recommend screening persons with epilepsy with an *O. volvulus* infection for glaucoma to prevent severe consequences such as blindness. In addition, there is a need to determine the prevalence of glaucoma among *O. volvulus*-infected persons without epilepsy and the effect of glaucoma treatment in *O. volvulus-*infected persons with and without epilepsy.

## Figures and Tables

**Figure 1 pathogens-11-01046-f001:**
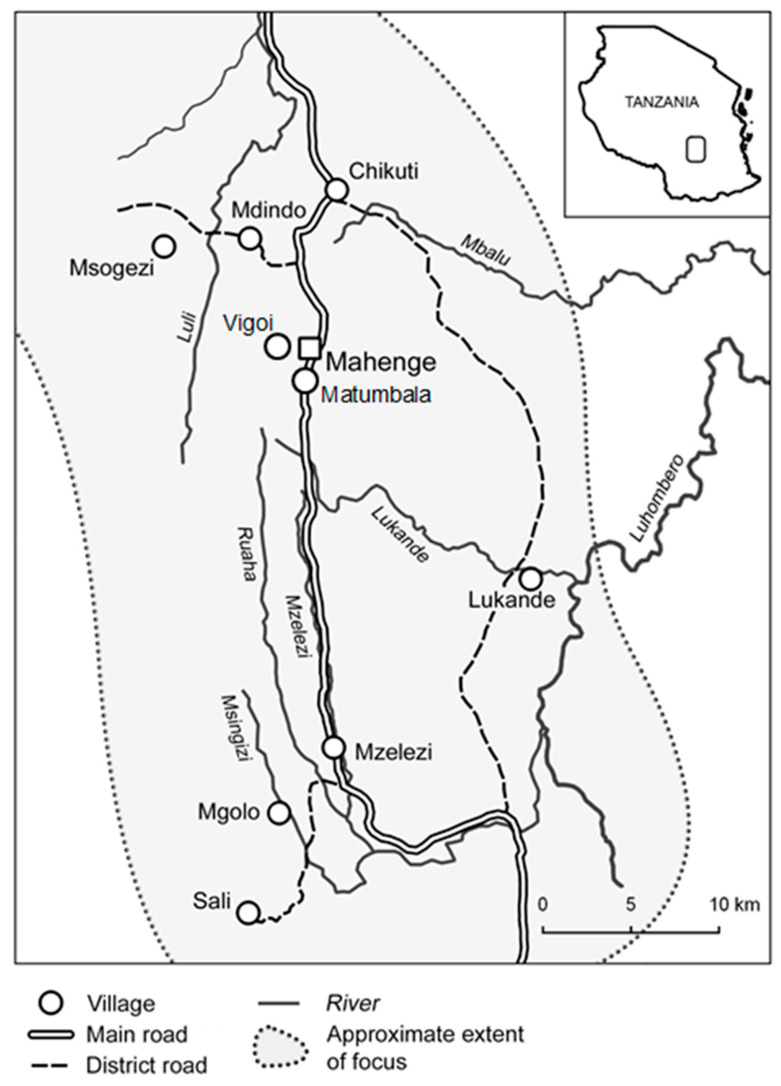
Ulanga district (Figure by A Hendy [7]).

**Figure 2 pathogens-11-01046-f002:**
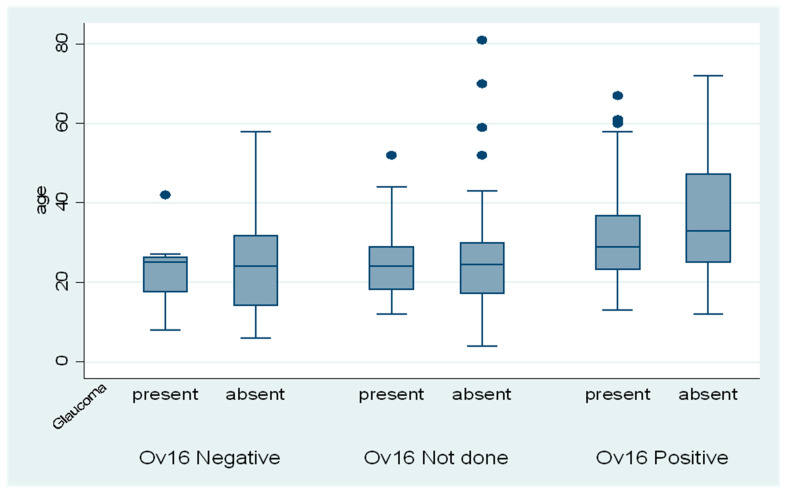
A box plot showing age (in years) distribution according to Ov16 test result (Positive/Negative/Not done) and the presence or absence of glaucoma among 278 persons with epilepsy.

**Table 1 pathogens-11-01046-t001:** Study population characteristics (n = 278).

Characteristic	Summary Measure
Age, median (IQR)	27 (21–38) years
Sex, n (%)	
Male	124 (44.6%)
Female	154 (55.4%)
Locality	
Sali	45 (15.2%)
Msogezi	44 (15.8%)
Mzelezi	41 (41.8%)
Mdindo	34 (12.2%)
Makanga	20 (7.2%)
Isongo	15 (5.4%)
Uponera	10 (3.6%)
Vigoi	10 (3.6%)
Nawenge	10 (3.6%)
Other	49 (17.6%)
Weight, mean (SD)	49 (11.9) kgs
Height, median(IQR)	153.5 (146–160) cm
Intraocular pressure, mean (SD)	
Right eye	18 (4.7) mmHg
Left eye	18 (4.6) mmHg
Epilepsy	
Nodding syndrome	50 (18.0%)
Other form of epilepsy	228 (82.0%)
Ov16 **	
Negative	59 (28.1%)
Positive	151 (71.9%)

IQR—Interquartile range; SD—Standard deviation; ** only 210 tested.

**Table 2 pathogens-11-01046-t002:** Ophthalmological findings (n = 278).

Diagnosis	Frequency (%)
Normal	145 (52.16)
Glaucoma	92 (33.09)
Vitreous opacities	18 (6.47)
Cataracts	8 (2.88)
Iris atrophy	6 (2.16)
Optic neuritis	2 (0.72)
Anterior uveitis	3 (1.08)
Optic atrophy	1 (0.36)
Chorioretinal atrophy	1 (0.36)
Others (burns, corneal scars)	2 (0.72)

**Table 3 pathogens-11-01046-t003:** The factors associated with onchocerciasis (n = 210).

Characteristic	Ov16 Negative n = 59	Ov16 Positive n = 151	Test Statistic	*p*-Value
Age, mean ± SD years	24.6 ± 11.7	33.9 ± 14.1	t = −4.829	<0.001
Body mass index,	20.3 ± 3.9	21.9 ± 10.4	t = −1.565	0.1194
kg/m^2^				
Sex			X^2^ = 0.1972	0.657
Male	27	64		
Female	32	87		
Epilepsy			X^2^ = 0.5208	0.471
Nodding syndrome	14	29		
Another form of epilepsy	44	119		
Glaucoma			X^2^ = 16.266	<0.001
Present	8	65		
Absent	51	86		

SD—Standard deviation.

**Table 4 pathogens-11-01046-t004:** Multivariate analysis of factors associated with onchocerciasis.

Characteristic	Crude IRR	*p*-Value	95% CI	Adjusted IRR	*p*-Value	95% CI
Glaucoma						
Absent	Reference		1	1		1
Present	1.58	<0.001	1.31–1.91	1.46	<0.001	1.24–1.70
Age	1.02	<0.001	1.01–1.02	1.01	<0.001	1.01–1.02
Sex						
Male	Reference		1			
Female	1.07	0.457	0.89–1.29			
Body mass index	1	0.07	0.99–1.01			
Epilepsy						
Another form of epilepsy	Reference		1			
Nodding syndrome	0.92	0.548	0.69–1.21			

IRR—Incidence rate ratio; 95% CI—95% Confidence interval.

## Data Availability

The datasets generated during and/or analysed during the current study are available from the corresponding author on reasonable request.

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
