# Peer review of "High Prevalence of Glaucoma among Patients in an Onchocerciasis Endemic Area (Mahenge, Tanzania)"

_pathogens, 2022, doi:10.3390/pathogens11091046_

Round 1
Reviewer 1 Report
In the current manuscript, the authors investigated the prevalence of glaucoma among epilepsy patients infected with Onchocerca volvulus. Onchocerciasis is a debilitating disease that can lead to severe dermatitis, vision impairment and even blindness. The pathology is induced by the death of the microfilariae, which reside in connective tissue such as the cornea of the eye and the skin. Thereby, the death of the microfilariae triggers an inflammation leading to neutrophil and eosinophil influx. This immune response results in tissue damage and thus into the conjunctivitis, sclerosing keratitis and glaucoma. Moreover, an increased frequency of epilepsy and nodding syndrome can be associated with onchocerciasis. In the current study, epilepsy patients living in Mahenge, Tanzania, which is an endemic area for onchocerciasis, were recruited and the prevalence of glaucoma and other ophthalmological findings were measured. An increased prevalence of glaucoma was demonstrated among Ov16 positive patients in comparison to Ov16 negative patients. In general, the manuscript is well written and the study is well conducted. However, some points should be addressed bevor publication.
Comments:
Materials and Methods:
2.1 Study site and population:
Is anything known about the prevalence of onchocerciasis in this region?
2.3 Study procedures:
What is the cross-reactivity of the Ov16 antigen test for other filarial diseases? Where other infections such as Trachoma investigated and ruled out?
Results:
Line 226: double Ov16. Delete last on Ov16.
Table 4: please delete the additional tab in the last row for column 2-4.
Discussion:
Line 242 ff: Can the authors speculate why there is an association between onchocerciasis infection and glaucoma in Ghana and Tanzania and not in South America?
Author Response
Reviewer 1
In the current manuscript, the authors investigated the prevalence of glaucoma among epilepsy patients infected with Onchocerca volvulus. Onchocerciasis is a debilitating disease that can lead to severe dermatitis, vision impairment and even blindness. The pathology is induced by the death of the microfilariae, which reside in connective tissue such as the cornea of the eye and the skin. Thereby, the death of the microfilariae triggers an inflammation leading to neutrophil and eosinophil influx. This immune response results in tissue damage and thus into the conjunctivitis, sclerosing keratitis and glaucoma. Moreover, an increased frequency of epilepsy and nodding syndrome can be associated with onchocerciasis. In the current study, epilepsy patients living in Mahenge, Tanzania, which is an endemic area for onchocerciasis, were recruited and the prevalence of glaucoma and other ophthalmological findings were measured. An increased prevalence of glaucoma was demonstrated among Ov16 positive patients in comparison to Ov16 negative patients. In general, the manuscript is well written and the study is well conducted. However, some points should be addressed bevor publication.
Comments:
Materials and Methods:
2.1 Study site and population:
Is anything known about the prevalence of onchocerciasis in this region?
Response
In 2017, the prevalence of O. volvulus antibodies among children 6-10 year old children in rural villages in the Mahenge area was 42.6% suggesting high ongoing onchocerciasis transmission. Based on this finding the onchocerciasis elimination programme in Mahenge was switched from annual to bi-annual CDTI.
Reviewer 1
2.3 Study procedures:
What is the cross-reactivity of the Ov16 antigen test for other filarial diseases? Where other infections such as Trachoma investigated and ruled out?
Response
The OV16 RTD test is considered to have low sensitivity but high specificity of > 90%. Cross -reactivity with other filarial infection is possible but in the Mahenge region Loa Loa , Lymphatic Filariasis and Mansonella perstans do not occur. Therefore it should be considered that all persons with epilepsy who were OV16 RDT positive had O. volvulus antibodies.
None of the study participants presented clinical signs of trachoma. Trachoma in the Mahenge area has been eliminated.
Reviewer 1
Results:
Line 226: double Ov16. Delete last on Ov16.
Response
We corrected this
Reviewer 1
Table 4: please delete the additional tab in the last row for column 2-4.
Response
We corrected. Epilepsy was analyzed as ‘Nodding syndrome’ and ‘Another form of epilepsy’ and this is now reflected as such in Table 4.
Reviewer 1
Discussion:
Line 242 ff: Can the authors speculate why there is an association between onchocerciasis infection and glaucoma in Ghana and Tanzania and not in South America?
Response
We added in the discussion:
“However, two individuals with microfilariae in the eye had intra-ocular pressure readings significantly above average. The apparent absence of glaucoma may be related to the small sample size but also to the lack of anterior eye inflammation because of limited ivermectin exposure. The difference this study and the African studies may also be related to difference in O volvulus strains between America and Africa.”
Botto C, Escalante A, Arango M, Yarzabal L. Morphological differences between Venezuelan and African microfilariae of Onchocerca volvulus. J Helminthol. 1988 Dec;62(4):345-51. doi: 10.1017/s0022149x00011755.
Choi YJ, Tyagi R, McNulty SN, Rosa BA, Ozersky P, Martin J, Hallsworth-Pepin K, Unnasch TR, Norice CT, Nutman TB, Weil GJ, Fischer PU, Mitreva M. Genomic diversity in Onchocerca volvulus and its Wolbachia endosymbiont. Nat Microbiol. 2016 Nov 21;2:16207. doi: 10.1038/nmicrobiol.2016.207
Reviewer 2 Report
The authors intended to test that having onchocerciasis may be a risk factor for developing raised intraocular pressure and glaucoma in the selected community, through measuring intraocular pressures and other ocular abnormalities, and determing the prevalence of glaucoma among persons with epilepsy attending a neurology clinic in Mahenge hospital and satellite treatment sites in rural villages in the area. Here, I have some minor comments for authors to be clarified. 1. 278 persons with epilepsy, including nodding syndrome, were selected in this study, but how many persons with the same syndrome in the whole selected areas? In other words, what percentage of 278? 2. Please correctly use the value of Chi-square test in the table 3. 3. Why 60 people didn't get tested for Ov16, and how will this affect subsequent analysis of the results? How is it assessed and dealt with? 4. It is suggested that the Discussion section be further enriched in response to the findings of the study.Author Response
Reviewer 2
The authors intended to test that having onchocerciasis may be a risk factor for developing raised intraocular pressure and glaucoma in the selected community, through measuring intraocular pressures and other ocular abnormalities, and determining the prevalence of glaucoma among persons with epilepsy attending a neurology clinic in Mahenge hospital and satellite treatment sites in rural villages in the area. Here, I have some minor comments for authors to be clarified. 1. 278 persons with epilepsy, including nodding syndrome, were selected in this study, but how many persons with the same syndrome in the whole selected areas? In other words, what percentage of 278?
Response
The epilepsy prevalence in the study area was 2.7%; 12.7% of the persons with epilepsy were classified as nodding syndrome. However, the total amount of persons with epilepsy in the Mahenge area is not known but is estimated to be at least more than 400.
Reviewer 2
- Please correctly use the value of Chi-square test in the table 3. 3.
Response
We corrected this by indicating that the Epilepsy was categorized as ‘Nodding syndrome’ and ‘another form of epilepsy’
Reviewer 2
Why 60 people didn't get tested for Ov16, and how will this affect subsequent analysis of the results? How is it assessed and dealt with?
Response
The people who were not tested for Ov16 antibodies were from the rural villages. We now mention this in the result section of the text.
Moreover in the discussion we state “A limitation of our study is also that 68 persons with epilepsy from the rural villages were not OV16 tested. However, because most persons with epilepsy visiting the epilepsy clinic at Mahenge hospital are also coming from rural villages we believe the study participants were representative of most persons with epilepsy in the area.”
Reviewer 2
- It is suggested that the Discussion section be further enriched in response to the findings of the study.
Response
We expanded the discussion
First we included a comment concerning the absence of glaucoma in the South American study
“However, two individuals with microfilariae in the eye had intra-ocular pressure readings significantly above average. The apparent absence of glaucoma may be related to the small sample size but also to the lack of anterior eye inflammation because of limited ivermectin exposure. The difference between this study and the African studies may also be related to strain differences between American and African O. volvulus strains.”
Botto C, Escalante A, Arango M, Yarzabal L. Morphological differences between Venezuelan and African microfilariae of Onchocerca volvulus. J Helminthol. 1988 Dec;62(4):345-51. doi: 10.1017/s0022149x00011755.
Choi YJ, Tyagi R, McNulty SN, Rosa BA, Ozersky P, Martin J, Hallsworth-Pepin K, Unnasch TR, Norice CT, Nutman TB, Weil GJ, Fischer PU, Mitreva M. Genomic diversity in Onchocerca volvulus and its Wolbachia endosymbiont. Nat Microbiol. 2016 Nov 21;2:16207. doi: 10.1038/nmicrobiol.2016.207
We also added more information about onchocerciasis and visual impairment reported in the literature
In Nord Kivu province, an onchocerciasis hyperendemic tropical rain forest area in the Democratic Republic of Congo (DRC), 39 (1.81%) out of 2150 subjects had onchocerciasis-related eye lesions and 4 (0.19%) were blind. Chorioretinitis (0.88%) was the most frequent onchocerciasis lesion followed by keratitis (0.46%), microfilaria in the anterior chamber (0.28%), iridocyclitis (0.28%), secondary glaucoma (0.19%), complicated cataract (0.19%), and optic atrophy (0.19%).
Baranwal VK, Shyamsundar K, Kabuyaya V, Biswas J, Vannadil H. Study of onchocerciasis-related visual impairment in North Kivu province of the Democratic Republic of Congo in Africa. Indian J Ophthalmol. 2020 May;68(5):890-894. doi: 10.4103/ijo.IJO_1653_18
CDTI has drastically reduced the threat of blindness in onchocerciasis endemic foci. This was shown in a study in Kaduna state, Nigeria, an onchocerciasis endemic area where after 27 years of CDTI, the prevalence of blindness dropped from 4.9 to 0.96%. In 2016, the most common causes of blindness were cataract (55.2%) and optic atrophy (27.6%), whereas the most common causes in 1989 were onchocerciasis (28.3%), glaucoma (17.4%) and cataract (10.9%). People with optic atrophy were more likely to have taken fewer doses of ivermectin over the years. In 2016, there was no case of anterior onchocerciasis seen among the visually handicapped, and only one individual (two eyes) with posterior onchocercal chorioretinitis. This contrasts with the finding in 1989 in which onchocerciasis, especially the posterior variety, was responsible for about one-third of the cases of blindness.
Babalola OE, Bassi A. Impact assessment study after 27 years of community-directed treatment with ivermectin in Galadimawa, Kaduna State, Nigeria. Niger Postgrad Med J. 2017 Jan-Mar;24(1):14-19. doi: 10.4103/npmj.npmj_6_17
It is still unclear whether certain O. volvulus strains may induce different ocular pathology. In a study in three ecological zones in Nigeria between 1998 and 1999, anterior segment onchocercal lesions, punctate and sclerosing keratitis were the predominant features of the infection in the savanna zone (14.1% and 6.3% respectively), while posterior segment lesions were much more common in the forest zone.
Umeh RE, Mahmoud AO, Hagan M, Wilson M, Okoye OI, Asana U, Biritwum R, Ogbu-Pearce P, Elhassan E, Yaméogo L, Braideo EI, Seketeli A. Prevalence and distribution of ocular onchocerciasis in three ecological zones in Nigeria.
Afr J Med Med Sci. 2010 Dec;39(4):267-75
It has been suggested that blindness is more prevalent in savannah than in forest onchocerciasis areas. However, a recent meta-analysis did not support the existence of a savannah blinding strain of onchocerciasis in West Africa.
Cheke RA, Little KE, Young S, Walker M, Basáñez MG. Taking the strain out of onchocerciasis? A reanalysis of blindness and transmission data does not support the existence of a savannah blinding strain of onchocerciasis in West Africa.
Adv Parasitol. 2021;112:1-50. doi: 10.1016/bs.apar.2021.01.002.
Round 2
Reviewer 1 Report
Only minor changes need to be addressed:
Line 88: male worms are 19-42 cm in length and not mm
Line 269 and 413: remove comma
Author Response
The male worms are much smaller than the female worms
We now mention they are 1.9-4.2 cm
We removed the commas
Reviewer 2 Report
Thank you for your reply. No more comments.
Author Response
The male worms are smaller than the female worms
We now mention they are 1.9-4.2cm in length
We removed the commas